# Social prescribing: less rhetoric and more reality. A systematic review of the evidence

Liz Bickerdike,[1] Alison Booth,[2] Paul M Wilson,[3] Kate Farley,[4] Kath Wright[1]

▶ Prepublication history and additional material is available. To view please visit the journal (http://dx.doi.org/10.1136/bmjopen-2016-013384).

For numbered affiliations see end of article.

**Correspondence to**
Paul Wilson; paul.wilson@manchester.ac.uk

## ABSTRACT

**Objectives:** Social prescribing is a way of linking patients in primary care with sources of support within the community to help improve their health and well-being. Social prescribing programmes are being widely promoted and adopted in the UK National Health Service and so we conducted a systematic review to assess the evidence for their effectiveness.

**Setting/data sources:** Nine databases were searched from 2000 to January 2016 for studies conducted in the UK. Relevant reports and guidelines, websites and reference lists of retrieved articles were scanned to identify additional studies. All the searches were restricted to English language only.

**Participants:** Systematic reviews and any published evaluation of programmes where patient referral was made from a primary care setting to a link worker or facilitator of social prescribing were eligible for inclusion. Risk of bias for included studies was undertaken independently by two reviewers and a narrative synthesis was performed.

**Primary and secondary outcome measures:** Primary outcomes of interest were any measures of health and well-being and/or usage of health services.

**Results:** We included a total of 15 evaluations of social prescribing programmes. Most were small scale and limited by poor design and reporting. All were rated as a having a high risk of bias. Common design issues included a lack of comparative controls, short follow-up durations, a lack of standardised and validated measuring tools, missing data and a failure to consider potential confounding factors. Despite clear methodological shortcomings, most evaluations presented positive conclusions.

**Conclusions:** Social prescribing is being widely advocated and implemented but current evidence fails to provide sufficient detail to judge either success or value for money. If social prescribing is to realise its potential, future evaluations must be comparative by design and consider when, by whom, for whom, how well and at what cost.

**Trial registration number:** PROSPERO Registration: CRD42015023501.

### Strengths and limitations of this study

- Social prescribing is a way of linking patients in primary care with sources of support within the community. It is being widely promoted and adopted as means of dealing with some of the pressures on general practice.
- This systematic review assesses the effectiveness of social prescribing programmes relevant to the UK National Health Service setting. We have searched for full publications and grey literature since 2000 and identified 15 evaluations. It is possible that some local evaluations have not been identified, but it is unlikely that any unidentified evaluations would do little to alter the overall picture of a low-quality evidence base with a high risk of bias.
- If social prescribing is to realise its potential, future evaluations must be comparative by design and consider when, for whom, how well and at what cost.

services are currently delivered remains high on the policy agenda. The Five Year Forward View has stressed that developing innovative approaches to delivering healthcare are integral to the long-term future of the National Health Service (NHS).[1]

Social prescribing is one such model and is being widely promoted as a way of making general practice (GP) more sustainable. Social prescribing is a way of linking patients in primary care with sources of support within the community. It provides GPs with a non-medical referral option that can operate alongside existing treatments to improve health and well-being. There is no widely agreed definition of social prescribing but the Social Prescribing Network defines it as 'enabling healthcare professionals to refer patients to a link worker, to co-design a non-clinical social prescription to improve their health and wellbeing.'[2] Schemes commonly use services provided by the voluntary and community sector and can include an extensive range of practical information and advice, community activity, physical activities,

## BACKGROUND

With estimates of a £30 billion funding gap by 2020, a radical rethink of the way health

befriending and enabling services. The types of activities offered as part of a social prescribing service can aim to help address the psychological problems and low levels of well-being often manifest in frequent attenders in GP. By addressing these, it is often hoped that there will be a subsequent positive impact on frequency of attendance.[3]

As early as 1999, the white paper *Saving Lives: Our Healthier Nation* was advocating that the NHS should make better use of community support structures and voluntary organisations.[4] However, it was in 2006 that the Department of Health advocated the introduction of social prescriptions for those with long-term conditions,[5] and NHS England has since announced the appointment of a national clinical champion for social prescribing.[6] With the current Secretary of State for Health also promoting access to non-clinical interventions that take a more 'holistic view',[1 7] support for social prescribing is significant at the policy level.

Many localities are now offering or considering implementing social prescribing programmes, but is the apparent enthusiasm justified? As part of a study which aimed to help NHS commissioners make better use of research in their decision-making,[8] we examined the evidence for social prescribing. This systematic review summarises the evidence for the effectiveness of social prescribing programmes relevant to the UK NHS setting.

## METHODS
The protocol and amendments were registered in PROSPERO (registration number: CRD42015023501).

### Data sources and searches
Database of Abstracts of Reviews of Effects (DARE), Cochrane Database of Systematic Reviews and NHS Economic Evaluation Database (NHS EED) were searched for relevant systematic reviews and economic evaluations (24 June 2015; no new records added to DARE and NHS EED databases from January 2015 so we did not run updated searches).

We searched the following databases (initial search 26 June 2015; updated search 5 February 2016): Applied Social Sciences Index and Abstracts (ASSIA), Cumulative Index to Nursing and Allied Health Literature (CINAHL), MEDLINE, Social Care Online and Social Policy and Practice.

As our focus was on identifying evidence relevant to the UK NHS setting, we also searched for eligible studies in key UK knowledge repositories for health and social care. The websites of National Institute for Health and Care Excellence (NICE), Social Care Institute for Excellence (SCIE) and NHS Evidence were searched for reviews, guidance, evidence briefings or any other papers describing or evaluating social prescribing programmes. Additional searches of the websites of key policy think tanks the Kings Fund, Health Foundation, Nuffield Trust and NESTA were also undertaken. We searched Google to identify grey literature reports of

relevant evaluations in UK settings (5 January 2016). Reference lists of retrieved articles were scanned to identify additional studies.

All the searches were restricted to English language only and published between 2000 to January 2016. The search strategies are available in online supplementary appendix 1.

### Study selection
Systematic reviews and any published evaluation of programmes where healthcare professionals refer patients from a primary care setting to a link worker or facilitator for any form of social prescription were eligible for inclusion. Studies were eligible regardless of whether a comparison group was included.

As per the Social Prescribing Network definition, we included only studies where referral was made from a primary care setting to a coordinator, link worker or facilitator of social prescribing (this type of role will be referred to as 'link workers' throughout this review). Any activities or interventions being specifically delivered as part of a social prescribing programme were included in the review.

We excluded studies where referral was made from outside of a primary care setting[9] and or where primary care health professionals refer patients to services delivered as part of mental health or counselling services such as an Improving Access to Psychological Therapies programme. We also excluded evaluations of activities that could be socially prescribed (eg, physical activity programmes or community arts projects) but did not involve referral to a link worker in the first instance.[10–13]

The primary outcomes of interest were any measures of health and well-being, including self-reported measures (eg, levels of physical activity or depression scores) and/or measure of usage of health services. We also considered any other outcomes (eg, health service usage) reported in the included evaluations.

Study selection was performed by one researcher and checked by a second, with any discrepancies resolved by discussion or with recourse to a third researcher.

### Data extraction and quality assessment
Details of the setting, participants, the intervention (type, delivery mode and length of time), type of evaluation and outcomes of evaluation were extracted and quality assessed by one researcher and checked by a second. Discrepancies were resolved by discussion or by recourse to a third researcher.

We used the Cochrane risk of bias tool to assess the quality of the randomised controlled trial (RCT).[14] To assess the quality of the before and after evaluations, we applied the quality assessment tool developed by the US National Heart, Lung, and Blood Institute for before–after (pre–post) studies with no control group.[15] Our primary focus was on effects. As per our protocol, we have not made a formal quality assessment of studies of a qualitative or descriptive nature.

## Data synthesis and analysis

We performed a narrative synthesis of the evidence. There were insufficient data to perform meta-analysis for any of the outcomes of interest. No subgroup analyses were planned. The narrative synthesis was intended to move beyond a preliminary summary of study findings and quality to investigate similarities and differences between studies as well as exploring any patterns in the data.

## RESULTS

We identified a total of 431 records through database searching and a further 14 records through other sources. After deduplication, 341 titles and abstracts were screened and 70 full-text papers were assessed for inclusion (see figure 1: Preferred Reporting Items for Systematic Reviews and Meta-Analyses (PRISMA) flow diagram).

## Excluded studies

We excluded 45 studies on eligibility grounds and were unable to access the full text for seven identified records. We also identified two non-systematic reviews of social prescribing schemes.[16] [17] These were excluded as they did not critically appraise included studies and were limited in their synthesis of findings; one review included a number of evaluations that did not meet our inclusion criteria.[17] We checked the reference lists of both reviews to ensure we had identified and included all relevant evaluations.

## Included studies

We included a total of 15 evaluations (reported in 16 papers) of social prescribing programmes where some form of link worker role was used.[3] [18–32] The designs included one RCT,[18] one non-RCT,[19] two qualitative studies,[23] [28] four uncontrolled before and after studies[3] [20–22] and eight descriptive reports of six evaluations, of which, five included some analysis of qualitative data.[24–27] [29–32] Details of the included evaluations are presented in table 1.

In each of the included studies, the link worker (job title variously named) met with the patient to discuss their needs and directed them to appropriate community/voluntary sector sources of support in their locality. The training and knowledge of people fulfilling these types of link worker role varied between projects. In some services this was a paid role, in others these roles were fulfilled by volunteers. Some link workers had good knowledge and existing networks with local services in place[28–30] and in others they received some basic training and made use of a directory of resources.[22]

Patients were referred to a range of activities provided by local or national voluntary and community sector organisations. Interventions received included exercise and other physical activities, signposting to housing,

**Figure 1** PRISMA flow diagram.

welfare and debt advice, adult education and literacy, befriending, counselling, self-help support groups, luncheon clubs and art activities.

The number of referrals made to social prescribing programmes ranged from 30–1607. Referrals were made by a range of health professionals but primarily GPs. Three of the studies reported that feedback was given to the referrer about the actions taken and the participants' progress in the social prescribing programme.[22 28 30]

### Quality of the evidence

Quality assessment and risk of bias for the evaluative designs is presented in table 2. In the RCT, only sequence generation was adjudged to be of low risk of bias; all other criteria were rated as unclear or high risk.[18] The authors reported that the randomisation process was misunderstood in two of the participating practices but random allocation appeared to be maintained. A key inclusion criterion for the Cochrane Effective Practice and Organisation of Care Review Group is that a controlled before and after study must have at least two interventions and two control groups to guard against confounding.[33] Here, the controlled before and after study includes one intervention and one control group, drawn from the same GP. As such, we rated the study as having a high risk of bias and made no further assessment of quality with the Cochrane risk of bias tool. Uncontrolled before and after studies are inherently weak evaluative designs and no included study fulfilled all of the specified quality criteria. In general, evaluations had small sample sizes (<100 participants), significant loss to follow-up (>20%), were lacking in completeness of outcome data and had unclear selection criteria for the study population. Follow-up periods were generally short (immediately post intervention, up to 4 months post intervention). There is therefore a high risk of bias.

### Uptake and attendance

Seven included studies reported the number of people attending an initial appointment with a link worker. Where reported, attendance at this initial appointment with a link worker ranged from 50% to 79%.[18 21–23 25–27] Participants' attendance at activities to which they were subsequently referred or recommended by a link worker was reported in only two studies and varied from 58%[22] to 100%.[21]

### Health and well-being outcomes

The RCT,[18] two uncontrolled before and after studies[21 22] and three descriptive reports[26 27 32] measured health and well-being outcomes at baseline and again at up to 6 months after referral to a social prescribing programme; one study reported outcomes at up to 12 months. The measures used were Warwick-Edinburgh Mental Well-being Scale (WEMWBS;[21 26 32]), Hospital Anxiety and Depression Scale (HADS;[18]),

General Anxiety Disorder-7 (GAD-7;[27]), Patient Health Questionnaire-9 (PHQ-9;[27]), Clinical Outcomes in Routine Evaluation-Outcome Measure (CORE-OM;[22]), Work and Social Adjustment Scale (WSAS;[21 22]), General Health Questionnaire (GHQ-12;[22]) and COOP/WONCA.[18] Table 3 presents findings for studies using validated measures; all report some improvements in health and well-being. However, it is difficult to quantify the size of the observed improvements due to a lack of reported detail, a lack of sufficient control group data and differences in reporting between studies. It is not possible to determine whether any observed improvements were clinically significant. Studies reported short-term outcomes only; there is no evidence about the effect social prescribing has on health and well-being outcomes beyond 6 months.

One uncontrolled before and after study used a bespoke measure, the Wellspring Well-being Questionnaire, comprising PHQ-9 and GAD-7 tools, and items from Office of National Statistics (ONS's) Well-being Index/Integrated Household Survey and International Physical Activity Questionnaires.[3] A second also used a bespoke measure which used a 5-point scale across eight domains associated with different aspects of self-management such as 'looking after yourself' and 'managing symptoms'.[20] Two further descriptive reports also indicated they used the WEMWBS to measure changes in health and well-being but are poor reported and involve what appear to be very small numbers of respondents.[24 25] In the two studies using non-validated measures, some positive improvements in outcomes such as depression and anxiety at 3–4 months' follow-up were reported.[3 20]

### Healthcare usage outcomes

Both comparative evaluations[18 19] and three uncontrolled before and after studies[3 20 22] reported some measure of healthcare usage. This included comparing hospital episode statistics and/or GP record data from 6 to 12 months before intervention with data up to 18 months post intervention. Reported outcomes included frequency of GP consultations, referrals to secondary care, inpatient admissions and accident and emergency (A&E) attendances. Findings were mixed. The RCT reported that the number of primary care contacts were similar between intervention and control groups and that there were fewer referrals to secondary care and more prescription drugs for those in the intervention group compared with the control group.[18] The non-randomised trial reported statistically non-significant reductions in primary care contacts (face-to-face and/or telephone) and referrals to secondary care.[19] The before and after studies reported reductions in secondary care referrals, inpatient admissions and A&E attendances[20] in primary care contact[22] and in face-to-face GP contact but an increase in telephone contact.[3]

**Table 1** Characteristics of social prescribing project evaluations

| Project information | Referral activity | Participants in evaluation (excluding health professionals and link workers) | Facilitator/coordinator skills and training | Activities patients referred to by social prescribing facilitator/coordinator |
|---|---|---|---|---|
| Project name, location: Amalthea project, Avon Author, year: Grant, 2000 Date project established (or time period of evaluation): Aug 1997 to Sep 1998 Type of evaluation: randomised controlled trial | Referred to link worker: N=90 Attended link worker appointment: 71/90 (79%) Attended a prescribed activity/ services: not reported GP surgeries involved: N=26 | Approached to participate: N=168 Agreed to participate: N=161 (90 randomised to intervention; 71 randomised to control) Participants in the control group received routine care from their GP Included in evaluation analysis: 69% of 90 for intervention an 67% of 71 for control followed up at 4 months | Three project facilitators from different backgrounds were trained and supervised by the organisation | Voluntary sector contacts available: National Schizophrenia Fellowship; Counselling on Alcohol and Drugs; Alcoholics Anonymous; Overeaters Anonymous; Local eating disorders group; Triumph over Phobia; Womankind; Counselling Network; CRUSE; RELATE; Befrienders International; Local carer support group; Princess Royal Trust for Carers; Royal British Legion; Crisis; Migraine Trust; Local assertiveness training group; National Society for the Prevention of Cruelty to Children; Multiple Sclerosis Society; Disability Living Foundation; British Trust for Conservation Volunteers; Citizens Advice Bureau; Local meet a mum association; local toddler group; local social group for the elderly; University of the Third Age; Brunelcare; Battle against tranquillisers; Women's Royal Voluntary Service |
| Project name, location: Connect project, Carlisle Author, year: Maughan, 2016 Date project established (or time period of evaluation):Oct 2011 to Mar 2014 Type of evaluation: controlled before and after study | Referred to link worker: not reportedAttended link worker appointment: N=30 Attended a prescribed activity/ services: not reported GP surgeries involved: N=1 | Approached to participate: not reported Agreed to participate: N=59 (30 in intervention group; 29 in control group) Participants in the control group received routine care from their GP Included in evaluation analysis: 28/30 (93%) in intervention; 29/ 29 (100%) in control | Non-healthcare staff, provided with brief training about local services, completing questionnaires and managing risk. Not reported | Available services across third, public and private sectors, self-help, self-management resources, educational, leisure and recreational facilities and fitness-, health- and exercise-related activities. Example given: The Eden Timebank, a skills exchange and social network where members earn credits for helping another member or the wider community. |

Continued

**Table 1** Continued

| Project information | Referral activity | Participants in evaluation (excluding health professionals and link workers) | Facilitator/coordinator skills and training | Activities patients referred to by social prescribing facilitator/ coordinator |
|---|---|---|---|---|
| Project name, location: Rotherham Social Prescribing project Author, year: Dayson, 2014 Date project established (or time period of evaluation): Apr 2012 to Mar 2014 Type of evaluation: uncontrolled before and after study | Referred to link worker: N=1607 Attended link worker appointment: not reported Attended a prescribed activity/ services: not reported (1118 people were referred onwards to other funded voluntary and community sector services) GP surgeries involved: N=29 | Approached to participate: not reported Agreed to participate: not reported Included in evaluation analysis: i. Hospital episode data analysis: N=451 followed up at 6 months; N=108 followed up at 12 months (of which n=42 referred on to a funded voluntary and community service provider) ii. Well-being outcomes analysis: 280/819 followed up at 3–4 months | Not reported. | Information and advice; community activity; physical activity; befriending and enabling |
| Project name, location: Dundee Equally Well Sources of Support Author, year: Friedli, 2012 Date project established (or time period of evaluation): Mar 2011 to Jun 2012 Type of evaluation: uncontrolled before and after study | Referred to link worker: N=123 Attended link worker appointment: 61/123 (50%) Attended a prescribed activity/ services: 26 out of 26 referred to an activity attended that activity (119 link worker referrals were made into 47 different community services or groups) GP surgeries involved: N=1 | Approached to participate: not reported Agreed to participate: not reported Included in evaluation analysis: N=16 | Not reported. | Community based information, support and/or activities |
| Project name, location: Graduate Primary Care Mental Health Worker Community Link Scheme, north London Author, year: Grayer, 2008 Date project established (or time period of evaluation):NR Type of evaluation: uncontrolled before and after study | Referred to link worker: N=255Attended link worker appointment: N=151 Attended a prescribed activity/ services: 58% attended at least one of the services suggested GP surgeries involved: N=13 | Approached to participate: N=151 Agreed to participate: 108/151 Included in evaluation analysis: N=75/108 followed up at 3 months | Psychology graduates with some voluntary clinical experience but no formal mental health training. Inhouse training and supervision from two clinical psychologists. Not reported. | Community resources identified through searches of paper and electronic directories, telephone enquiries and other sources. |

| Project information | Referral activity | Participants in evaluation (excluding health professionals and link workers) | Facilitator/coordinator skills and training | Activities patients referred to by social prescribing facilitator/ coordinator |
|---|---|---|---|---|
| Project name, location: Well-being Programme at Wellspring Healthy Living Centre, Bristol<br>Author, year: Kimberlee, 2014<br>Date project established (or time period of evaluation): May 2012 to Apr 2013<br>Type of evaluation: uncontrolled before and after study | Referred to link worker: Unclear<br>Attended link worker appointment: N=128<br>Attended a prescribed activity/ services: not reported<br>GP surgeries involved: not reported | Approached to participate: N=128<br>Agreed to participate: N=128<br>Included in evaluation analysis:<br>i. Health and well-being outcomes N=70 followed up at 3 months<br>ii. GP attendance data N=40 12 months before and after baseline | Not reported | Peer support groups, creative arts, physical activities, cooking courses, complementary therapies |
| Project name, location: Age Concern, Yorkshire and Humber<br>Author, year: Age Concern, 2012<br>Date project established (or time period of evaluation): Apr 2011 to Sep 2011<br>Type of evaluation: descriptive report | Referred to link worker: N=55<br>Attended link worker appointment: not reported<br>Attended a prescribed activity/ services: not reported<br>GP surgeries involved: N=12 | Approached to participate: unclear<br>Agreed to participate: unclear<br>Included in evaluation analysis: not reported | A skilled member of Age UK staff | Age UK services including: befriending, day clubs, luncheon clubs, information and advice, benefit checks, trips, theatre outings, computer training, advocacy, legal advice, will-writing service, volunteering, Fit as a Fiddle classes, art groups, memory loss services |
| Project name, location: ConnectWell, Coventry<br>Author, year: Baines, 2015<br>Date project established (or time period of evaluation): Aug 2014 to Aug 2015<br>Type of evaluation: descriptive report (with qualitative element) | Referred to link worker: N=39<br>Attended link worker appointment: 24/39 (62%)<br>Attended a prescribed activity/ services: not reported<br>GP surgeries involved: N=4 | Approached to participate: not reported<br>Agreed to participate: not reported<br>Included in evaluation analysis: N=5 | Volunteers attend group training session then inductions for specific role. Additional training offered eg, mentoring, dementia awareness. Supervised by WCAVA | Befriending, lunch club, advice and information services, housing/ homelessness services, counselling, sport, art, volunteering, support group, social activities |
| Project name, location: Newcastle Social Prescribing Project<br>Author, year: ERS Research and Consultancy, 2013 | Referred to link worker: N=124<br>Attended link worker appointment: 87/124 (70%)<br>Attended a prescribed activity/ services: not reported<br>GP surgeries involved: N=6 | Approached to participate: not reported<br>Agreed to participate: not reported<br>Included in evaluation analysis: N=9 | Existing staff member in each VCSO with knowledge of local community and services, LTCs. Skills and attributes specified. | Support with personalised goal setting and buddying, self-care and signposting to information, advice and support through an agency: Age UK; HealthWORKS; |

**Table 1** Continued

| Project information | Referral activity | Participants in evaluation (excluding health professionals and link workers) | Facilitator/coordinator skills and training | Activities patients referred to by social prescribing facilitator/coordinator |
|---|---|---|---|---|
| Involve North East, 2013 Date project established (or time period of evaluation): Jan 2012 to Mar 2013 Type of evaluation: two descriptive reports (one with qualitative element) | | | | Newcastle Carers; Search; West End Befrienders |
| Project name, location: CHAT, south and west Bradford Author, year: Woodall, 2005 Date project established (or time period of evaluation): Established 2004 Piloted Jan 2005 to Sep 2005 Type of evaluation: descriptive report (with qualitative element) | Referred to link worker: N=81 Attended link worker appointment: not reported Attended a prescribed activity/services: not reported GP surgeries involved: N=3 | Approached to participate: not reported Agreed to participate: not reported Included in evaluation analysis: N=10 | Non-clinical Health Trainers, a public health workforce supported by the DH | Local community and voluntary services. |
| Project name, location: CHAT, south and west Bradford Author, year: South, 2008 Date project established (or time period of evaluation): May 2005 to Oct 2006 Type of evaluation: qualitative study | Referred to link worker: N=223 Attended link worker appointment: not reported Attended a prescribed activity/services: not reported GP surgeries involved: not reported | Approached to participate: not reported Agreed to participate: not reported Included in evaluation analysis: N=10 | Non-clinical Health Trainers, a public health workforce supported by the DH | Community and voluntary sector groups and services such as: Luncheon clubs; Befriending groups; Social services; Volunteering organisations; Getting back into work groups; Literacy classes; Debt advice; Access bus; Bereavement groups; Reminiscing groups; Arts and craft groups; Music groups |
| Project name, location: Health Trainer and Social Prescribing Service, south and west Bradford Author, year: White 2010 Date project established (or time period of evaluation): Established 2006 (evolved from CHAT) Jan 2010 to Sep 2010 | Referred to link worker: N=484 Attended link worker appointment: not reported Attended a prescribed activity/services: not reported GP surgeries involved: N=21 | Approached to participate: not reported Agreed to participate: not reported Included in evaluation analysis: N=12 | Non-clinical Health Trainers, a public health workforce supported by the DH | Local voluntary and community sector social groups and support agencies. Health trainer can develop personal health action plan. |

**Table 1** Continued

| Project information | Referral activity | Participants in evaluation (excluding health professionals and link workers) | Facilitator/coordinator skills and training | Activities patients referred to by social prescribing facilitator/ coordinator |
|---|---|---|---|---|
| Type of evaluation: descriptive report (with qualitative element) Project name, location: Doncaster Patient Support Service Author, year: Faulkner, 2004 Date project established (or time period of evaluation): April 2001 to February 2002 Type of evaluation: qualitative study | Referred to link worker: 200 Attended link worker appointment: N=132 Attended a prescribed activity/ services: not reported GP surgeries involved: N=1 | Approached to participate: 17 patients and 9 volunteers Agreed to participate: Patients: N=11 Volunteers: N=9 Included in evaluation analysis: Patients: N=11 Volunteers: N=9 | Volunteers given 3-day training including basic counselling knowledge and skills, team building strategies and visits from community services they might refer people to. Ongoing training and supervision provided. | Facilitated access to services providing: advice on disability services, advice on nursing homes; alcohol support; benefit issues; family/matrimonial support; family support for drug users; advice on housing/social services; legal issues (eg, The Women's Centre; Mind; Relate; Alcohol and Drug Advice) |
| Project name, location: WellFamily service in Hackney* Author, year: Longwill, 2014 Date project established (or time period of evaluation): First established 1996; Period of evaluation: 2012–13 Type of evaluation: descriptive report (with qualitative element) | Referred to link worker: N=1466 Attended link worker appointment: N=1089 Attended a prescribed activity/ services:N=712 GP surgeries involved: N=32 | Approached to participate: not reported Agreed to participate:Not reported Included in evaluation analysis: GAD-7, PHQ-9: N=387 Patient survey: N=92 respondents (out of active caseload of ~120) GP survey: N=27 respondents (out of 160 surveyed GPs) | Family action workers and senior practitioners with a variety of skills and experience. Some with undergraduate and postgraduate qualifications in counselling, group therapy, medicine and psychotherapy. Family Action counsellors— professionally qualified and under regular supervision | Short-term counselling, advice and practical support. Local voluntary, community, and social enterprise sector services. Other social and health services such as debt counselling, housing departments and health services |
| Project name, location: 'New Routes', Keynsham (Bath and North East Somerset) Author, year: Brandling, 2011 Date project established (or time period of evaluation):2-year pilot established October 2009 Type of evaluation: descriptive report (with qualitative element) | Referred to link worker: N=90 Attended link worker appointment: not reported Attended a prescribed activity/ services: N=42 GP surgeries involved: N=3 | Approached to participate: Not reported Agreed to participate: Not reported Included in evaluation analysis: WEMWBS completed at 6– 12 months N=7 MYMOP2 completed at 6– 12 months N=12 Qualitative interviews N=21 | Coordinators role modelled on Amalthea project[13] Skills and training not reported | 46 different types of organisations and activities were part of the pilot. Most popular activities: volunteering; befriending; walking groups; art groups |

CHAT, Community Health Advice Team; DH, Department of Health; GP, general practice; GAD-7, General Anxiety Disorder-7; LTC, Long Term Condition; MYMOP2, Measure Yourself Medical Outcome Profile; NR, not reported; PHQ-9, Patient Health Questionnaire-9; VCSO, Voluntary Community Sector Organisation; WCAVA, Warwickshire Community and Voluntary Action; WEMWBS, Warwick Edinburgh Mental Well-being Scale.

**Table 2** Quality assessment and risk of bias

**Uncontrolled before and after evaluations**

| Study | Quality criteria | Risk of bias | Notes |
|---|---|---|---|
| **Grant 2000 RCT** | Sequence generation | Low | Sealed opaque envelopes prepared by research team. Stratification by practice and blocks of six used (three intervention/three control). |
| | Allocation concealment | Unclear | Sequentially numbered envelopes opened. In two practices, there was evidence that the randomisation process was initially misunderstood: six patients excluded. |
| | Blinding of participants and personal | Not possible | |
| | Blinding of outcome assessment | Unclear | |
| | Incomplete outcome data | High | 32% loss to follow-up at 4 months |
| | Selective outcome reporting | Unclear | |
| | Other potential threats to validity | Unclear | Numbers potentially eligible but not recruited unknown Recruited general practices were not a random sample: participating doctors were likely to be more interested in the research question and may have managed psychosocial problems more actively, which could have diminished reported estimates of effects |
| **Maughan 2016 CBA** | Is there a suitable comparison group? | Yes | One intervention and one control group, drawn from the same general practice with similar patient characteristics. |
| | Do the authors use theory to underpin the project/evaluation? | No | |
| | Were appropriate methods used for data collection and analysis? | Yes | Models environmental costs (in terms of carbon footprint) |
| | Were efforts made to assess patient experience? | No | Data were retrospectively collected from GP health records for a 2-year period. Two participants in intervention group excluded from analysis Financial and environmental impacts calculated for each outcome using national averages or accepted conversion factors |

**Uncontrolled before and after evaluations**

| Study | Quality criteria | Risk of bias | Notes |
|---|---|---|---|
| **Dayson 2014** | Was the study question or objective clearly stated? | Yes | Small sample of those referred (N=1607) participated in evaluation— HES data at 6 months N=451, at 12 months N=108; well-being data at 3–4 months 280/819 |
| | Were eligibility/selection criteria for the study population prespecified and clearly described? | Not reported | |
| | Were the participants in the study representative of those who would be eligible for the test/service/intervention in the general or clinical population of interest? | Yes | Methods of qualitative analysis of patient experience unclear |
| | Were all eligible participants that met the prespecified entry criteria enrolled? | Not reported | |
| | Was the sample size sufficiently large to provide confidence in the findings? | No | |
| | Was the test/service/intervention clearly described and delivered consistently across the study population? | Not reported | |
| | Were the outcome measures prespecified, clearly defined, valid, reliable and assessed consistently across all study participants? | Yes | |
| | Were the people assessing the outcomes blinded to the participants' exposures/interventions? | Not reported | |

Continued

**Table 2** Continued

**Uncontrolled before and after evaluations**

| Study | Quality criteria | Risk of bias | Notes |
|-------|------------------|--------------|-------|
| | Was the loss to follow-up after baseline 20% or less? Were those lost to follow-up accounted for in the analysis? | No | |
| | Did the statistical methods examine changes in outcome measures from before to after the intervention? Were statistical tests done that provided p values for the pre-to-post changes? | Yes | |
| | Were outcome measures of interest taken multiple times before the intervention and multiple times after the intervention (ie, did they use an interrupted time-series design)? | No | |
| | If the intervention was conducted at a group level (eg, a whole hospital, a community, etc), did the statistical analysis take into account the use of individual-level data to determine effects at the group level? | Not applicable | |
| Friedli 2012 | Was the study question or objective clearly stated? | Yes | Details of preintervention and postintervention outcomes not reported |
| | Were eligibility/selection criteria for the study population prespecified and clearly described? | No | Small sample size |
| | | | Timing of post intervention assessment not reported |
| | Were the participants in the study representative of those who would be eligible for the test/service/intervention in the general or clinical population of interest? | Yes | Methods of qualitative analysis of patient and provider/referrer experience unclear |
| | Were all eligible participants that met the prespecified entry criteria enrolled? | Not applicable | |
| | Was the sample size sufficiently large to provide confidence in the findings? | No | |
| | Was the test/service/intervention clearly described and delivered consistently across the study population? | Not reported | |
| | Were the outcome measures prespecified, clearly defined, valid, reliable and assessed consistently across all study participants? | No | |
| | Were the people assessing the outcomes blinded to the participants' exposures/interventions? | No | |
| | Was the loss to follow-up after baseline 20% or less? Were those lost to follow-up accounted for in the analysis? | No | |
| | Did the statistical methods examine changes in outcome measures from before to after the intervention? Were statistical tests done that provided p values for the pre-to-post changes? | No | |
| | Were outcome measures of interest taken multiple times before the intervention and multiple times after the intervention (i.e., did they use an interrupted time-series design)? | No | |
| | If the intervention was conducted at a group level (eg, a whole hospital, a community, etc), did the statistical analysis take into account the use of individual-level data to determine effects at the group level? | Not applicable | |

Continued

**Table 2** Continued

**Uncontrolled before and after evaluations**

| Study | Quality criteria | Risk of bias | Notes |
|---|---|---|---|
| **Grayer 2008** | Was the study question or objective clearly stated? | Yes | GP practices volunteered and may not be representative of practices overall |
| | Were eligibility/selection criteria for the study population prespecified and clearly described? | Yes | Patients who consented to participate in evaluation were more likely to speak English as a first language than those who did not consent |
| | Were the participants in the study representative of those who would be eligible for the test/service/intervention in the general or clinical population of interest? | Yes | No significant differences at baseline between those successfully followed up and those lost to follow-up |
| | Were all eligible participants that met the prespecified entry criteria enrolled? | No | 95% CIs (no p values) reported for changes in GHQ-12, CORE-OM and WSAS scores |
| | Was the sample size sufficiently large to provide confidence in the findings? | No | |
| | Was the test/service/intervention clearly described and delivered consistently across the study population? | Yes | |
| | Were the outcome measures prespecified, clearly defined, valid, reliable and assessed consistently across all study participants? | Yes | |
| | Were the people assessing the outcomes blinded to the participants' exposures/interventions? | Not reported | |
| | Was the loss to follow-up after baseline 20% or less? Were those lost to follow-up accounted for in the analysis? | No | |
| | Did the statistical methods examine changes in outcome measures from before to after the intervention? Were statistical tests done that provided p values for the pre-to-post changes? | Yes | |
| | Were outcome measures of interest taken multiple times before the intervention and multiple times after the intervention (i.e., did they use an interrupted time-series design)? | No | |
| | If the intervention was conducted at a group level (eg. a whole hospital, a community, etc), did the statistical analysis take into account the use of individual-level data to determine effects at the group level? | Not applicable | |
| **Kimberlee 2014** | Was the study question or objective clearly stated? | Yes | SROI analysis presents data for all baseline completers and the smaller percentage who were followed up; possible bias towards positive finding for intervention |
| | Were eligibility/selection criteria for the study population prespecified and clearly described? | No | |
| | Were the participants in the study representative of those who would be eligible for the test/service/intervention in the general or clinical population of interest? | Yes | Unclear whether calculations of mean differences in scale scores used all baseline data or baseline data for follow-up completers only |
| | Were all eligible participants that met the prespecified entry criteria enrolled? | Not applicable | p values reported for change from baseline at 3 months in PHQ-9 depression scores |
| | Was the sample size sufficiently large to provide confidence in the findings? | No | |
| | Was the test/service/intervention clearly described and delivered consistently across the study population? | Not reported | |
| | | Yes | |

Continued

**Table 2** Continued

**Uncontrolled before and after evaluations**

| Study | Quality criteria | Risk of bias | Notes |
|---|---|---|---|
| | Were the outcome measures prespecified, clearly defined, valid, reliable and assessed consistently across all study participants? | Not reported | |
| | Were the people assessing the outcomes blinded to the participants' exposures/interventions? | No | |
| | Was the loss to follow-up after baseline 20% or less? Were those lost to follow-up accounted for in the analysis? | No | |
| | Did the statistical methods examine changes in outcome measures from before to after the intervention? Were statistical tests done that provided p values for the pre-to-post changes? | Yes | |
| | Were outcome measures of interest taken multiple times before the intervention and multiple times after the intervention (i.e., did they use an interrupted time-series design)? | No | |
| | If the intervention was conducted at a group level (eg. a whole hospital, a community, etc), did the statistical analysis take into account the use of individual-level data to determine effects at the group level? | Not applicable | |

CBA, Controlled Before and After study; CORE-OM, Clinical Outcomes in Routine Evaluation-Outcome Measure; GP, general practice; GHQ, General Health Questionnaire; HES, hospital episode statistics; PHQ, Patient Health Questionnaire; RCT, randomised controlled trial; SROI, Social Return on Investment; WSAS, Work and Social Adjustment Scale.

## Patient experience

Three before and after studies[20–22] and five descriptive reports[23 26 28 30 32] reported on patient experience. Studies used semistructured interviews or survey questionnaires specifically designed for the project evaluation to assess participant experience.

In six of the studies, participants reported overall satisfaction with social prescribing programmes.[20–22 26 28 30] General improvements in feelings of loneliness and social isolation,[21 30 32] and improved mental and physical health were also observed.[21] Issues that may impact the willingness of patients to participate in socially prescribed activities included confidence,[21 30] interest in/appropriateness of activities on offer[21 30] and literacy or travel issues.[30 32] One qualitative study reported that patients had poor knowledge of the service prior to attending their appointment with the link worker resulting in some feeling that the service did not meet their expectations.[23] Another evaluation identified a similar issue regarding a lack of understanding of the service among participants.[32]

## Referrer experience and lessons learnt

A small number of studies conducted semistructured interviews with primary care practitioners referring participants to social prescribing programmes and/or link workers.[21 26 28–32] GPs in general found that being able to make a social prescription was a useful additional tool.[21 28 29 31] Key issues identified for successful implementation of social prescribing programmes were central coordination of referrals,[26] resources and training to support coordinators and enabling networking with the voluntary and community sector,[26 29] and good communication between GPs, participants and link workers: social prescribing is unfamiliar to many GPs and requires good clear explanation to engage participants;[21 23 26 32] delivering feedback on participants' progress encourages GP support for social prescribing.[28 30 31]

## Costs

The two comparative evaluations reported costs. One found total mean costs were greater in the intervention group (£153) compared with the control group (£133).[18] The other reported no statistically significant differences between the financial and environmental costs of healthcare use between the intervention and control groups.[19]

One before and after study undertook a cost-benefit analysis using estimated input costs and benefits derived from 12-month outcome data obtained for 108 patients referred to social prescribing (42 of whom were referred to funded voluntary and community service providers). A total NHS cost reduction of £552 189 was generated by multiplying the estimated per-patient cost reduction by the total number of referrals (n=1118) to funded voluntary and community service providers over the 2-year course of the pilot programme. This estimate was compared with total estimated input costs of £1.1 million.[20]

**Table 3** Health and well-being outcomes (validated measures)

| Study (timing of outcome measurement post baseline measurement) | WEMWBS | HADS | GAD-7 | PHQ-9 | CORE-OM | WSAS | GHQ-12 | COOP/WONCA |
|---|---|---|---|---|---|---|---|---|
| RCTs | | | | | | | | |
| Grant 2000 (4 months) | | Intervention group (N=62) * greater improvement than control group (N=48)* | | | | | | Intervention group (N=62) * greater improvement than control group (N=48) * |
| Before and after evaluations | | | | | | | | |
| Friedli 2012 (NR) | 'Statistically significant improvement' in mental well-being (N=16) (scores not reported) | | | | | 'Statistically significant improvement' in functional ability (N=16)(scores not reported) | | |
| Grayer 2008 (3 months) | | | | | Small reduction in patients categorised as cases (N=74) | Improvement in work and social adjustment (N=69) | Four-fifths were cases at baseline, reducing to half of post intervention N=69) | |
| Descriptive reports | | | | | | | | |
| ERS Research and Consultancy 2013 (NR) | Increase in mean score from 22 to 26 (N=16) | | | | | | | |
| Longwill 2014 (NR) | | | 2.5 points reduction in score (p<0.001) (N=387) | 3.1 points reduction in score (p<0.001) (N=387) | | | | |
| Brandling 2011 (6–12 months) | 'General positive trend but owing to low number of participants completing questionnaires no further conclusions can be made' | | | | | | | |

*calculated from reported percentage followed up at 4 months.
COOP/WONCA, Dartmouth COOP Functional Health Assessment Charts; CORE-OM, Core Outcome Measure; GAD-7, General Anxiety Disorder-7; GHQ-12, General Health Questionnaire-12; HADS, Hospital Anxiety and Depression Scale; PHQ-9, Patient Health Questionnaire-9; RCT, randomised controlled trial; WEMWBS, Warwick-Edinburgh Mental Well-being Scale; WSAS, Work and Social Adjustment Scale.

One other report of an evaluation estimated total running costs of £83 144 for the programme for 1 year.[3]

## DISCUSSION

This systematic review has examined the evidence to inform the commissioning of social prescribing schemes. Overall, we identified 15 evaluations conducted in UK settings but have found little convincing evidence for either effectiveness or value for money.

Most of the evaluations of social prescribing activity are small scale and limited by poor design and reporting. Missing information has made it difficult to assess who received what, for what duration, with what effect and at what cost. Common design weaknesses include a lack of comparators (increasing the risk of bias), loss to follow-up, short follow-up durations and a lack of standardised and validated measuring tools. There is also a distinct failure to either consider and/or adjust for potential confounding factors, undermining the ability to attribute any reported positive outcomes to the intervention (or indeed interventions) received. This is particularly important as most referred patients appear to have been receiving other interventions and so we have no way of assessing the relative contributions of the interventions to the outcomes reported. Despite these methodological shortcomings, most evaluations have presented positive conclusions, generating a momentum for social prescribing that does not appear to be warranted.

### Strengths and limitations

Our systematic review appears to be the first to assess the effectiveness of social prescribing programmes relevant to the UK NHS setting. We have searched for full publications and grey literature since 2000 but it is possible that we have not identified some local evaluations. Publication bias occurs when the results of published studies are systematically different from results of unpublished studies. However, we think it unlikely that any unidentified evaluations will be more robust than those included in the review.

Many of the evaluations presenting positive conclusions were written as descriptive reports with limited or no supporting data presented. As such, they did not adhere to formal reporting standards that would be expected in reports to funding agencies or in academic journal articles. This made extracting any relevant data difficult and it is possible information relevant to outcomes is missed. Even if this shortcoming of data completeness were to be addressed, we believe that it would do little to alter the overall picture of a low-quality evidence base with a high risk of bias.

### Implications

Our systematic review has not established that there is clear evidence that social prescribing is ineffective. Rather, we are not yet able to reliably judge which, if any, social prescribing programmes demonstrate a degree of promise and so could be considered further. The use of a link worker is the key feature of social prescribing. How this link worker role was fulfilled varied significantly between projects. So here again, we are not able to reliably judge the type of skills set or level of training and knowledge people require to effectively fulfil this role. For those seeking to commission new or extend existing schemes, this evidence gap is a hindrance rather than a help, especially so given the widespread support and advocacy for social prescribing at the policy level.

While the tension between rigour and 'good enough' evidence has long been recognised,[34] even 'good enough' is severely lacking from the social prescribing literature be that in the design or in the conduct of the evaluations themselves. This may in part reflect the way schemes have 'emerged' rather than being systematically planned with evaluation built in from the outset. Nevertheless, if social prescribing is to realise its potential, then there is an urgent need to improve the ways by which schemes are evaluated.

Prospective pathways for undertaking rigorous planned experimental evaluation are well defined,[35] but the opportunity, time and resources needed to employ these in a service context can be limited. However, this does not serve as an excuse for inaction and in the current financial climate we should of course only be investing in those services where we can demonstrate real benefit over existing ways of working. What this should mean for future evaluation of social prescribing is that a more coordinated approach to the planning, implementation and evaluation of new and existing schemes is undertaken. This could and should involve the adoption of a common analytical framework which in turn will facilitate standardised metrics, cross-site comparison and shared learning. The IDEAL framework offers one such pathway to navigate the evaluation continuum that would allow for the iterative development and evaluation of whether social prescribing is likely to succeed in a particular setting and allow for adaptation, refinement and system integration without losing sight of the need for more rigorous testing before wider spread.[36] Whatever analytical framework is adopted, Lamont and colleagues[37] have proposed five essential questions for evaluation which those planning to undertake evaluations of social prescribing programmes would do well to heed. These are:

▶ Why—clarify aims and establish what we already know from evidence.
▶ Who—identify and engage stakeholders and likely users of research at outset.
▶ How—think about study design, using an appropriate mix of methods, and adjust for bias where possible (or at least acknowledge).
▶ What—consider what to measure (activity, costs and outcomes) and combine data from different sources.

▸ When—pay attention to timing of results to maximise impact.

Alongside these, we would also emphasise that rigorous conduct and transparent reporting (regardless of 'success' or 'failure') are essential. Reporting guidelines such as Standards for QUality Improvement Reporting Excellence (SQUIRE)[38] with its focus on explaining 'Why did you start?', 'What did you do?', 'What did you find?' and 'What does it mean?' could readily be applied to ensure that learning is systematically captured in a generalisable format. This in turn would serve to ensure that any future decisions relating to the continuation or wider spread of social prescribing schemes are transparent and evidence informed.

## CONCLUSIONS

Social prescribing is being widely advocated and implemented but current evidence fails to provide sufficient detail to judge either success or value for money. If social prescribing is to realise its potential, future evaluations must be comparative by design and consider when, by whom, for whom, how well and at what cost.

**Author affiliations**
[1]Centre for Reviews and Dissemination, University of York, York, UK
[2]York Trials Unit, University of York, York, UK
[3]Alliance Manchester Business School, University of Manchester, Manchester, UK
[4]School of Healthcare, University of Leeds, Leeds, UK

**Contributors** PMW took overall responsibility for the systematic review. LB, AB and PMW were involved in all stages of the review from development of the protocol, through screening studies and data extraction, to analysis and synthesis and production of the final manuscript. KF provided input at all stages of the review and commented on drafts of the review. KW conducted literature searches and contributed to the 'Methods' section of the review. All authors approved the final version and PMW is the guarantor.

**Funding** This review was funded by the National Institute for Health Research (NIHR). As part of research funded by the NIHR Health Services and Delivery Research programme (Project ref: 12/5002/18), Additional funding for PMW was received from the NIHR Collaboration for Leadership in Applied Health Research and Care (CLAHRC) Greater Manchester.

**Disclaimer** The views expressed are those of the authors and do not necessarily reflect those of the NIHR Health Services and Delivery Research programme, NIHR CLAHRC Greater Manchester or the Department of Health.

**Competing interests** None declared.

**Provenance and peer review** Not commissioned; externally peer reviewed.

**Data sharing statement** All available data can be obtained from the corresponding author.

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
