## [Reviewer comments · BMJ Open]

ARTICLE DETAILS

TITLE (PROVISIONAL)	Social prescribing: less rhetoric and more reality. A systematic review of the evidence
AUTHORS	Bickerdike, Liz; Booth, Alison; Wilson, Paul; Farley, Kate; Wright, Kath

VERSION 1 - REVIEW

REVIEWER	Marieke Kroezen KU Leuven Institute for Healthcare Policy, Catholic University Leuven
REVIEW RETURNED	17-Jul-2016

GENERAL COMMENTS	Overall comment: This is a well-written article with clear policy relevance. The methods used are well described and fit the aims of the paper. Below I provide some minor comments and suggestions, but overall I feel this article is very much worthy of publication. Minor comments per section: Abstract: - Could you please define 'social prescribing' in the abstract? For a non-UK reader, it is not clear from the start what this entails.- Conclusions: I would add "(..) and consider when, by whom, for whom, how well and at what cost." Background I was wondering whether social prescribing, or something similar, already happens in other countries and whether some lessons may be learned from that? If not, perhaps it would be good to state that this is a very UK-context specific phenomenon. Methods The search strings seem fine, although I cannot judge how integrated the term 'social prescribing' is in the UK (research) and whether perhaps alternative terms with the same meaning should have been included. Results - When reading the results, I was sometimes a bit confused what 'participation in a social prescribing programme' exactly means. Do patients already fall under this scope if they only saw a link-worker? Or do patients have to see a link-worker and be referred to further interventions (and perhaps participate in them) to fall under the scope of this definition?
---

	 - Perhaps it should be made more clear what the 'control group' in some of the original studies was: patients who were not referred to a link worker but did receive a referral to a further intervention from a GP? - Also, for almost all outcomes discussed, especially the health and wellbeing outcomes, I find it very hard to distinguish the social prescribing-results from the results which the interventions patients were referred to achieved? In other words, is a decrease in BMI attributable to social prescribing or to a loose weight-intervention (that the link-worker referred the patient to)? Perhaps in the discussion some remarks could be made about how well studies could distinguish these results. Discussion/implications/conclusions Like I already mentioned for the 'Abstract', I would add some remarks on the role of the link worker itself. So future evaluations should consider "(..) when, by whom, for whom, how well and at what cost." After all, if I look at table 1, the facilitators/coordinators differ considerably, ranging from young voluntary psychology graduates to senior practitioners. This may have considerable influence on how the social prescriber role is filled in and hence, the results that it achieves. Some remarks should be included about this.
--	---

REVIEWER	Molly Courtenay Cardiff University, UK
REVIEW RETURNED	10-Aug-2016

GENERAL COMMENTS	A very clear and comprehensive review. Researchers undertaking work in this area would find the review very helpful.
--

REVIEWER	Karen Mattick University of Exeter, UK
REVIEW RETURNED	26-Sep-2016

GENERAL COMMENTS	This is an important topic and the authors make a strong argument for the need for the research. I hope the following comments will help the authors to strengthen their manuscript still further. Abstract I think it will be important to justify the decision to not look beyond the UK, particularly given the relatively small number of included papers. I think the case for this can be made but it not explicit in the paper currently. I would avoid making a conclusion about value for money, given that this aspect has not been referred to as an aim of the study or mentioned through the previous sections. Methods I thought it was helpful to present the PRISMA guidance in Appendix 2 but this did serve to highlight some aspects of the manuscript that might be developed further. For example, it may be helpful to be explicit in terms of the components of your review with respect to
---

	PICO (item 4). More detail in some sections of the methods is required to provide the rationale for various decisions taken (item 6). For example, how did you decide which databases and websites to search? Why did you select the time period 2000-2016? Under study selection, you will need to state that you were only looking at UK settings and justify this. You will also need to give the rationale for excluding evaluations that did not involve referral to a link-worker in the first instance. Under data extraction, I think more explanation is required for how you assessed the quality of those included studies that were not RCTs or before and after designs – and why you used the US National Heart, Lung and Blood Institute tool for the latter, rather than others you might have chosen. A comment about the extent to which the evaluation of ‘risk of bias’ can be considered a quality assessment more generally is needed – and also how you used the findings of this analysis to influence the conclusions drawn. Much more detail is also required about how the narrative synthesis was undertaken. Results In this section, Table 1 provides a very useful summary of the data – in my print out, the text in the final column was truncated, so I was unable to review it – but the common subheadings under ‘referral activity’ and ‘participants’ work well. In Figure 1, the reasons for excluding studies at each stage need to be presented (PRISMA item 17). On p8 line 16 I think there is a typo – g should be groups? Again it will be important in this section to comment on the quality of the other included students that were not RCTs or B&A studies. On p8 lines 49-56, the formatting has gone awry in relation to the brackets, references, colons and semi-colons – please be consistent with this. Again the subheading of costs within this section comes as something of a surprise to the reader since this is not pre-empted in the aims statement, although cost is mentioned at the end of the Strengths & Limitations section. I think this could be ‘set up’ a little better? Although research ethics is unlikely to have been required for this study, an explicit statement on this would be helpful. Discussion It will be important to stipulate in the first paragraph that you have found 15 evaluations in the UK. Publication bias is not mentioned in the limitations section (PRISMA item 15). The last part of this section provides helpful advice for future studies.
--	---

REVIEWER	Kathleen Rice Dalla Lana School of Public Health/Institute for Health Policy, Management, and Evaluation University of Toronto Toronto, ON, Canada
REVIEW RETURNED	10-Oct-2016

GENERAL COMMENTS	This manuscript presents the findings and implications of a systematic review of social prescribing programmes relevant to the UK NHS setting. Is an important and timely manuscript that warrants publication, provided a few minor issues are addressed. The study objectives, methods, and writing is clear overall, with the necessary conceptual and theoretical armature well-defined throughout.
---

Given BMJ publishes studies carried out outside the UK, I think it would be advisable to specify in the opening sentence of the manuscript that you are writing specifically about the UK. This becomes obvious given you go on to discuss the NHS and the Five Year Forward View, but in its current form it is initially unclear whose funding gap and policy agenda is being discussed.

Please explain "enabling" (under Background heading, paragraph 2).

Under the Study Selection heading, I would suggest adding a sentence that briefly summarizes what is meant by "formal evaluations." Please also add a sentence explaining what "narrative synthesis" means, under the Data Synthesis and analysis heading. Given this review will be of interest to policymakers as well as researchers, I think it would be unwise to assume that all interested readers will be familiar with this concept.

Under the Strengths and Limitations heading: authors claim that "many of the evaluations are written as narrative reports and as such do not adhere to formal reporting standards that would be expected in reports to funding agencies or in academic journal articles." I would strongly suggest you specify the kind of journals you are talking about (e.g. medicine-oriented journals, perhaps?). I assume that this statement reflects the authors' expertise as health-focused researchers, but this statement is not accurate as written. Many journals across a range of disciplines publish articles that are written in a narrative style. Also, it is possible that the authors found it difficult to extract relevant data from articles written in a narrative style because their discipline-specific training has not emphasized extracting information from articles that are written in this style (in other words, the authors may have less experience extracting information from these sorts of documents). This could perhaps be addressed in this paragraph.

Spelling and Grammar:

-Line 7 of the abstract should read: "...some of the pressures on general practice by supporting people's access...". There is currently an 's missing between people and access.

P.g. 4, Line 21: In the interest of clarity, I suggest changing the wording to read "...patients to link a worker, who will co-design..."

Pg. 4, Line 40: missing period at the end of the paragraph.

Second paragraph under the Included Studies heading: missing S after "link-worker"

First paragraph under Quality of Evidence paragraph (page 8, line 15): should read "...two control group" ; word "group" is missing.

The second sentence under the Uptake and attendance heading (pg. 8, lines 35, 36) is a sentence fragment, and its meaning is unclear.

Pg. 9 lines 20-23: sentence fragment, I think there is a word missing. Consequently, the meaning is unclear.

P.g. 10 lined 58: extra period at the end of the paragraph.

	Page 11, line 51: I find the first sentence under the Implications heading to be unclear as written, and suggest something along these lines: "Our systematic review has not established that social prescribing is ineffective." Overall, this is an important review, and I recommend its publication provided these minor points are addressed.
--	--

VERSION 1 – AUTHOR RESPONSE

Reviewer 1	
Comments	Responses
Abstract: -Could you please define 'social prescribing' in the abstract? For a non-UK reader, it is not clear from the start what this entails. -Conclusions: I would add "(..) and consider when, by whom, for whom, how well and at what cost."	We have added a definition to the abstract Thank you for highlighting this point; we have added 'by whom' to the sentence
Background I was wondering whether social prescribing, or something similar, already happens in other countries and whether some lessons may be learned from that? If not, perhaps it would be good to state that this is a very UK-context specific phenomenon.	Social prescribing appears to be a recent phenomenon and very UK specific. We have amended the background section to make this UK policy context clearer.
Methods The search strings seem fine, although I cannot judge how integrated the term 'social prescribing' is in the UK (research) and whether perhaps alternative terms with the same meaning should have been included.	Our search strategies were designed by a senior information specialist. The term social prescribing appears well integrated and the search strings also utilised terms such as community and non-medical referral to broaden the search for potentially relevant studies.
Results -When reading the results, I was sometimes a bit confused what 'participation in a social prescribing programme' exactly means. Do patients already fall under this scope if they only saw a link-worker? Or do patients have to see a link-worker and be referred to further interventions (and perhaps participate in them) to fall under the scope of this definition?	Our use of participation lacks clarity and we now have reworded as 'referral to a social prescribing programme' which is a more accurate description (see p8).

-Perhaps it should be made more clear what the 'control group' in some of the original studies was: patients who were not referred to a link worker but did receive a referral to a further intervention from a GP? -Also, for almost all outcomes discussed, especially the health and wellbeing outcomes, I find it very hard to distinguish the social prescribing-results from the results which the interventions patients were referred to achieved? In other words, is a decrease in BMI attributable to social prescribing or to a loose weight-intervention (that the link-worker referred the patient to)? Perhaps in the discussion some remarks could be made about how well studies could distinguish these results.	Only 2 studies had 'control' groups. In both instances, patients in the control group received routine care from their GP. This information has been added to Table 1. You find it hard to distinguish the outcomes for social prescribing because it is hard. The lack of consideration of and adjustment for confounding variables is a key feature of this evidence base, and undermines its value for decision making purposes. We have mentioned, confounding in the discussion but have added additional reflection on p11 as this is a key take home message.
Discussion/implications/conclusions Like I already mentioned for the 'Abstract', I would add some remarks on the role of the link worker itself. So future evaluations should consider "(..) when, by whom, for whom, how well and at what cost." After all, if I look at table 1, the facilitators/coordinators differ considerably, ranging from young voluntary psychology graduates to senior practitioners. This may have considerable influence on how the social prescriber role is filled in and hence, the results that it achieves. Some remarks should be included about this.	As above, we have added 'by whom' to the sentence in question and added some additional reflection on the apparent variation in expertise and role of the link-worker (see p12).

Reviewer 2	
A very clear and comprehensive review. Researchers undertaking work in this area would find the review very helpful.	Thank you for your comments – no response required

Reviewer 3	
Abstract I think it will be important to justify the decision to not look beyond the UK, particularly given the relatively small number of included papers. I think the case for this can be made but it not explicit in the paper currently. I would avoid making a conclusion about value for money, given that this aspect has not been referred to as an aim of the study or mentioned through the previous sections.	Our inclusion criteria of 'any formal evaluations' implicitly includes cost studies. We have amended to make this more explicit
Methods	

I thought it was helpful to present the PRISMA guidance in Appendix 2 but this did serve to highlight some aspects of the manuscript that might be developed further. For example, it may be helpful to be explicit in terms of the components of your review with respect to PICO (item 4). More detail in some sections of the methods is required to provide the rationale for various decisions taken (item 6). For example, how did you decide which databases and websites to search? Why did you select the time period 2000-2016? Under study selection, you will need to state that you were only looking at UK settings and justify this. You will also need to give the rationale for excluding evaluations that did not involve referral to a link-worker in the first instance. Under data extraction, I think more explanation is required for how you assessed the quality of those included studies that were not RCTs or before and after designs – and why you used the US National Heart, Lung and Blood Institute tool for the latter, rather than others you might have chosen.	We address the signposting to specific PRISMA elements below: The PICO components are already stated but we have reworded and re-ordered the study selection section to enhance the signposting of the individual components (see p5-6). We selected what is a standard range of health and social care databases to search for eligible studies. We have reworded the section covering the UK centric data sources and searches (see p5). Social prescribing is a recent phenomenon and our search time period reflects this. Although as early as 1999, the white paper Saving Lives: Our Healthier Nation was advocating that the NHS should make better use of community support structures and voluntary organisations, it is not until 2006 that the notion of social prescribing is first advocated. We have amended the background section to make this policy context clearer. The rationale for focusing on the UK setting is already presented in the introduction. We have added additional reminders to our focus throughout the manuscript As we state in the introduction, the use of a link worker is the key feature of social prescribing. We have therefore made the Social Prescribing Networks definition more explicit to add clarity (see p4). Our primary focus was on effects. As per our protocol we have not made a formal quality assessment of studies of a qualitative or descriptive nature. Uncontrolled before and after designs are an inherently weak design with a high risk of bias, even when well conducted. Because of this, they are rarely included in reviews assessing effects. Cochrane actively discourage their inclusion in reviews. The recently developed ROBINS-I tool does not extend to UBA designs. We anticipated the potential of a low quality evidence base but wanted to present comprehensive summary of the available evidence. In doing so, we used the US NHLBI tool as it is widely used and its descriptive qualities are recognised.
--	---

A comment about the extent to which the evaluation of 'risk of bias' can be considered a quality assessment more generally is needed – and also how you used the findings of this analysis to influence the conclusions drawn. Much more detail is also required about how the narrative synthesis was undertaken.	'Risk of bias' is otherwise known as quality assessment. We believe we are already very explicit in how we have used our finding that the overriding characteristic of the evidence base is a high risk of bias. This is why the focus of our discussion is on the need for higher quality evidence. The narrative synthesis was intended to move beyond a preliminary summary of study findings and quality to investigate similarities and differences between studies as well as exploring any patterns in the data. However, the overriding finding is that of poorly conducted evaluations with a high risk of bias. This therefore is the focus of our discussion.
Results In this section, Table 1 provides a very useful summary of the data – in my print out, the text in the final column was truncated, so I was unable to review it – but the common subheadings under 'referral activity' and 'participants' work well. In Figure 1, the reasons for excluding studies at each stage need to be presented (PRISMA item 17). On p8 line 16 I think there is a typo – g should be groups? Again it will be important in this section to comment on the quality of the other included studies that were not RCTs or B&A studies. On p8 lines 49-56, the formatting has gone awry in relation to the brackets, references, colons and semi-colons – please be consistent with this. Again the subheading of costs within this section comes as something of a surprise to the reader since this is not pre-empted in the aims statement, although cost is mentioned at the end of the Strengths & Limitations section. I think this could be 'set up' a little better? Although research ethics is unlikely to have been required for this study, an explicit statement on this would be helpful.	We have reformatted all the tables to fit. Reasons for exclusion at each stage are provided in Fig 1. Yes typo and now amended We have added - Our primary focus was on effects. As per our protocol we have not made a formal quality assessment of studies of a qualitative or descriptive nature Amended for consistency Our inclusion criteria of 'any formal evaluations' implicitly includes cost studies. To make this more explicit we have now reworded to state '...any formal evaluations (i.e. quantitative, qualitative and or costs) of social prescribing programmes...' to the section on study selection (p5). Research ethics is not required for systematic reviews. No further response is necessary.
Discussion It will be important to stipulate in the first paragraph that you have found 15 evaluations in the UK. Publication bias is not mentioned in the	Amended We have already mentioned publication bias but

limitations section (PRISMA item 15). The last part of this section provides helpful advice for future studies.	not explicitly – the section is now amended to read: We have searched for full publications and grey literature since 2000 but it is possible that we have not identified some local evaluations. Publication bias occurs when the results of published studies are systematically different from results of unpublished studies. However, we think it unlikely that any unidentified evaluations will be more robust than those included in the review.
--	--

Reviewer 4	
Given BMJ publishes studies carried out outside the UK, I think it would be advisable to specify in the opening sentence of the manuscript that you are writing specifically about the UK. This becomes obvious given you go on to discuss the NHS and the Five Year Forward View, but in its current form it is initially unclear whose funding gap and policy agenda is being discussed.	We have made the rationale for focusing on the UK setting more explicit (see p4).
Please explain "enabling" (under Background heading, paragraph 2).	The word 'services' should have been included – now reads as 'enabling services'
Under the Study Selection heading, I would suggest adding a sentence that briefly summarizes what is meant by "formal evaluations." Please also add a sentence explaining what "narrative synthesis" means, under the Data Synthesis and analysis heading. Given this review will be of interest to policymakers as well as researchers, I think it would be unwise to assume that all interested readers will be familiar with this concept.	We have reworded this section as use of formal is unhelpful – by formal we really meant published. We have amended to state any published evaluation We added the following explanation: “The narrative synthesis was intended to move beyond a preliminary summary of study findings and quality to investigate similarities and differences between studies as well as exploring any patterns in the data.”
Under the Strengths and Limitations heading: authors claim that "many of the evaluations are written as narrative reports and as such do not adhere to formal reporting standards that would be expected in reports to funding agencies or in academic journal articles." I would strongly suggest you specify the kind of journals you are talking about (e.g. medicine-oriented journals, perhaps?). I assume that this statement reflect the authors' expertise as health-focused researchers, but this statement is not accurate as written. Many journals across a range of disciplines publish articles that are written in a narrative style. Also, it is possible that the authors found it difficult to extract relevant data from articles written in a narrative style because their discipline-specific training has not emphasized extracting information from articles that are written in this style (in other words, the authors	The reviewer has misunderstood the point we were trying to make. It is not that we lack expertise in extracting information from articles written in narrative style, rather the reports are descriptive and present limited (i.e only a graph or figure) or indeed no data to support positive conclusions. It is therefore possible that relevant information is missing from these reports. We have reworded the paragraph on p11 to make this issue clearer: 'Many of the evaluations presenting positive conclusions were written as descriptive reports with limited or no supporting data presented. As such, they did not adhere to formal reporting standards that would be expected in reports to funding agencies or in academic journal articles. This made extracting any relevant data difficult

may have less experience extracting information from these sorts of documents). This could perhaps be addressed in this paragraph.	and it is possible information relevant to outcomes has been missed. Even if this shortcoming of data completeness were to be addressed we believe that it would do little to alter the overall picture of a low quality evidence base at high risk of bias.'
Spelling and Grammar: -Line 7 of the abstract should read: "...some of the pressures on general practice by supporting people's access...". There is currently an 's missing between people and access. P.g. 4, Line 21: In the interest of clarify, I suggest changing the wording to read "...patients to link a worker, who will co-design..." Pg. 4, Line 40: missing period at the end of the paragraph. Second paragraph under the Included Studies heading: missing S after "link-worker" First paragraph under Quality of Evidence paragraph (page 8, line 15): should read "...two control group" ; word "group" is missing. The second sentence under the Uptake and attendance heading (pg. 8, lines 35, 36) is a sentence fragment, and its meaning is unclear. Pg. 9 lines 20-23: sentence fragment, I think there is a word missing. Consequently, the meaning is unclear. P.g. 10 lined 58: extra period at the end of the paragraph. Page 11, line 51: I find the first sentence under the Implications heading to be unclear as written, and suggest something along these lines: "Our systematic review has not established that social prescribing is ineffective."	Amended Amended Added Added Added Sentence rephrased as follows: "Where reported, attendance at this initial appointment with a link worker ranged from 50% to 79%." Sentences in the Health care utilisation section have now been rephrased for clarity. Added Amended as suggested
Overall, this is an important review, and I recommend its publication provided these minor points are addressed.	Thank you

VERSION 2 – REVIEW

REVIEWER	Kathleen Rice Dalla Lana School of Public Health/Institute of Health Policy, Management, and Evaluation University of Toronto Toronto, Canada
REVIEW RETURNED	06-Dec-2016

GENERAL COMMENTS	This version is much improved. It is an important and timely systematic review, and I recommend its publication.
--